# Bilinguals are better than monolinguals in detecting manipulative discourse

**Evelina Leivada**[1,2]*, **Natalia Mitrofanova**[2], **Marit Westergaard**[2,3]

**1** Universitat Rovira i Virgili, Tarragona, Spain, **2** UiT-The Arctic University of Norway, Tromsø, Norway, **3** NTNU Norwegian University of Science and Technology, Trondheim, Norway

* evelina.leivada@urv.cat

**Data Availability Statement:** The data are held in the following public repository, hosted by Universitat Rovira i Virgili: http://repositori.urv.cat/fourrepopublic/search/item/PC%3A3577.

## Abstract

One of the most contentious topics in cognitive science concerns the impact of bilingualism on cognitive functions and neural resources. Research on executive functions has shown that bilinguals often perform better than monolinguals in tasks that require monitoring and inhibiting automatic responses. The robustness of this effect is a matter of an ongoing debate, with both sides approaching bilingual cognition mainly through measuring abilities that fall outside the core domain of language processing. However, the mental juggling that bilinguals perform daily involves language. This study takes a novel path to bilingual cognition by comparing the performance of monolinguals and bilinguals in a timed task that features a special category of stimulus, which has the peculiar ability to manipulate the cognitive parser into treating it as well-formed while it is not: grammatical illusions. The results reveal that bilinguals outperform monolinguals in detecting illusions, but they are also slower across the board in judging the stimuli, illusory or not. We capture this trade-off by proposing the *Plurilingual Adaptive Trade-off Hypothesis* (PATH), according to which the adaptation of bilinguals' cognitive abilities may (i) decrease fallibility to illusions by means of recruiting sharpened top-down control processes, but (ii) this is part of a larger bundle of effects, not all of which are necessarily advantageous.

## Introduction

Speaking more than one language has been argued to confer certain anatomical and behavioral effects. At the behavioral level, these effects have often been referred to with the term 'bilingual advantage', which describes bilinguals/plurilinguals as (i) performing better in executive control tasks that require the inhibition of goal-irrelevant distractors [1], (ii) showing a larger benefit from cues in inhibitory control [2], (iii) returning faster to a baseline state after competitor inhibition [3], and (iv) exhibiting a greater task-switching flexibility [4]. However, not all relevant experiments find evidence for such effects, as executive function (EF) tasks have also provided results that suggest that bilinguals do not outperform monolinguals on the relevant measures [5–7]. Importantly, both sides of the debate are predicated on one crucial assumption: bilingual language control recruits general executive control. In other words, the status of the bilingual advantage as a robust phenomenon is controversial [8], but what is not controversial is the domain in which the advantage is standardly searched for: EFs.

**Funding:** This work received support from the European Union's Horizon 2020 research and innovation programme under the Marie Skłodowska-Curie grant agreement n˚ 746652 (to EL) and from the Spanish Ministry of Science, Innovation and Universities under the Ramón y Cajal grant agreement n˚ RYC2018-025456-I (to EL). The publication charges for this article have been funded by a grant from the publication fund of UiT The Arctic University of Norway. The funders had no role in study design, data collection and analysis, decision to publish, or preparation of the manuscript.

**Competing interests:** The authors have declared that no competing interests exist.

Several factors may explain the existence of contradictory evidence for and against bilingual effects in EFs. One possible explanation is that there are inherent issues related to task granularity in EFs [9]. The results from the neuroanatomical front are more homogeneous and indicate robust brain changes after exposure to another language [10, 11]. Moreover, the notion of domain-general inhibitory control as a unitary psychometric construct has been recently questioned [12], and this suggests that wholesale effects, observed uniformly across all EF tasks, are not to be expected. Another parameter to consider is that even though bilingualism is about managing, inhibiting and monitoring *language*, studies on bilingual effects often use EFs as their testing vehicle. When linguistic tasks are employed, these often measure lexical fluency, not grammar [13]. Research in fluency has produced findings that suggest a bilingual disadvantage [14]: Bilinguals score lower than monolinguals on verbal fluency [15], are slower in picture naming [16], and encounter more tip-of-the-tongue experiences [17].

Observing that the occurrence of bilingual effects in EFs is not ubiquitous, one might consider other domains of testing to find such effects. This proposal stems from the following question: why should the experience of bilinguals in inhibiting or switching language translate into faster reaction times in EF tasks that measure conflict-resolution or the ability to ignore salient, distracting cues? As Treccani & Mulatti put it, we take for granted that there is a relationship between the processes exercised when bilinguals handle two languages and the processes that underlie other cognitive functions, *without* having a solid theory about this relationship [18]. In the absence of such a theory and in appreciation of the fact that the cognitive effort of bilinguals concerns *language*, we follow a long line of literature in looking for bilingual effects in the domain of language processing itself [15–17], and more specifically, in a type of linguistic stimulus that has not been tested from this perspective so far: linguistic illusions. The background hypothesis is that if the mental juggling inherent to the constant activation and suppression of two languages translates into pronounced cognitive aptitude (defined as the synergistic outcome of mental abilities such as speed of computation, mental flexibility, attention orienting, etc. [19]), different degrees of cognitive aptitude may lead to differential performance in tasks that aim to trick the cognitive parser, especially if these tasks use *linguistic devices* in order to manipulate the addressee.

It is well-established that the parser can be led into making incorrect judgments, because it operates on the basis of certain processing heuristics [20]. These incorrect decisions that systematically deviate from the reality and are hard to suppress (i.e., they arise automatically) are often referred to as illusions [21]. For example, the Müller-Lyer illusion is a classical visual illusion, in which the length of a line is judged to be longer when its ends are followed by arrows pointing inward than when its ends are followed by arrows pointing outward (Fig 1) [22].

Even if one is familiar with this illusion and knows for a fact that the length of the two lines is equal, one still cannot avoid automatically perceiving the first line as longer. Similarly, in the linguistic domain, illusions may trick the parser into giving an incorrect answer (1) or into accepting an uninformative, ill-formed, and meaningless sentence as informative, well-formed, and meaningful (2).

1. How many animals of each kind did Moses take into the arc? [23]

2. More people have been to Russia than I have. [24]

Sentences like (1) show that our parser does not always check and suppress intuitive answers. Instead of constructing a complete representation of the properties of this utterance, a processing heuristic that settles on a "good enough" representation and produces an automatic answer is employed [25]. In this case, the automatic answer is two, but the correct answer is none, because Moses did not take any animal into the arc; Noah did [23]. Similarly,

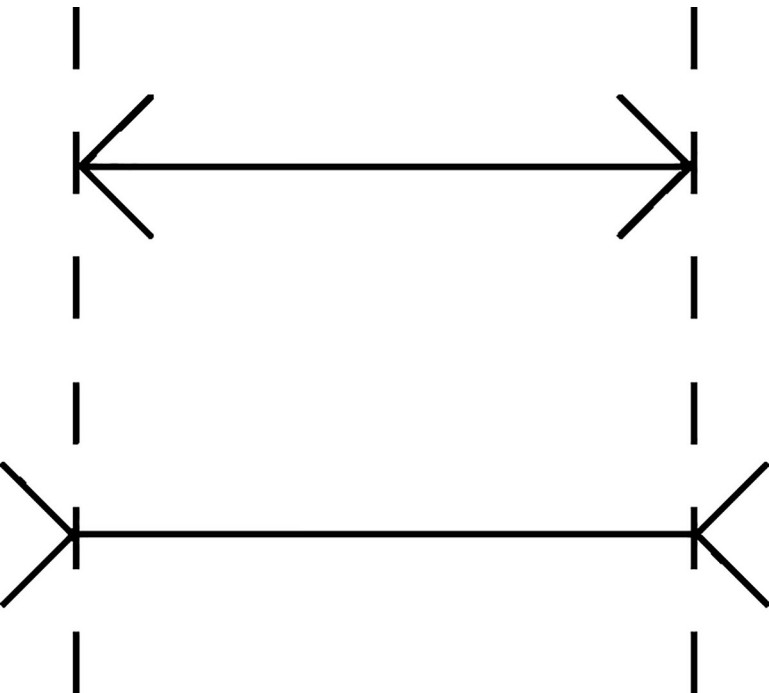

**Fig 1. The Müller-Lyer illusion.**

(2) is standardly given high acceptability judgments [26, 27], but when people who have accepted it as well-formed are asked to provide its meaning, they are often surprised to find out that this task is unexpectedly hard. This happens because the superficial well-formedness of (2) is an illusion: the sentence involves a comparison of *individuals* in the matrix clause and a comparison of *events* in the embedded clause. In linguistic terms, (2) is ill-formed because the matrix subject 'more people' calls for a comparison of individuals, but the embedded subject 'I' is not a bare plural (cf. the grammatically well-formed 'More people have been to Russia than koala have'), so there is no appropriate host for the comparison [28]. Sentence (1) is known as the Moses illusion, while (2) is an example of the comparative illusion.

The illusions just mentioned (i.e., Müller-Lyer, Moses, and comparative) have two things in common: (i) they use a processing heuristic to elicit an automatic answer, because (ii) the parser recognizes that the employed heuristic is usually beneficial for processing in terms of a reasonable least effort-best result trade-off. More specifically, the heuristic in (1) is that the information is pragmatically relevant and semantically correct [23], in (2) it is the presupposition that the utterance has a meaning [29], and in the Müller-Lyer illusion that size is calculated through an averaging process that is sensitive to different cues and experiences, such that the heuristic is employed by cultures that live in highly carpentered urban environments [30, 31]. In all three illusions, *context* is of paramount importance. If Moses is replaced by Nixon, which is a name that does not appear in a biblical context, the illusion in (1) disappears [23]. If the context is altered such that the arrows are presented separately before the line, the Müller-Lyer illusion disappears and the effect that emerges goes in the opposite direction (i.e., the line that temporally succeeds the inward arrows is found to be shorter) [32]. The emergence of the comparative illusion also relies heavily on contextual triggers in the sense that the illusion arises within a manipulative context that constrains the search of a meaning through specific devices [29, 33]. In the linguistic illusions literature, manipulation has been described as a covert speech act that aims to mislead the context selection process when a person interprets a

target utterance that features an illusion [33]. More specifically, through the use of phonological, syntactic, semantic, and pragmatic devices, such as the use of the wrong name in (1), albeit one that arises in the appropriate context and has the appropriate length, or the mix of two locally meaningful comparatives in (2), the process of providing an answer to illusions is manipulated to elicit an automatic, biased response. Fallibility to linguistic illusions can thus be understood as the outcome of shallow processing that operates through employing processing heuristics as well as general communicative principles (e.g., the principle of relevance) [25, 29, 34]. Overall, linguistic illusions trick the cognitive parser in a way similar to how attention and awareness are manipulated in magicians' performances with optical illusions: they provide and partition information in a way that tricks the addressee to perceive a stimulus in a way that does not match the physical reality of the stimulus [35].

An unexplored way of approaching susceptibility to such manipulation is by testing whether people with different degrees of cognitive aptitude are more able to suppress intuitive answers and do the processing required to detect illusions. The underlying connection we seek to establish between bilingual cognition and cognitive illusions boils down to the fact that the magnitude of the latter can be influenced by top-down control processes [36]. For instance, the magnitude of the Müller-Lyer illusion is affected by voluntary attentional disengagement and orienting to the arrows [37]. We observe a link between (i) language-switching, as a factor that may drive the superior performance of bilinguals in EF tasks by means of recruiting top-down control mechanisms, which are sharpened by switching from one language to another in order to address different monolingual interlocutors [38], and (ii) the recruitment of certain control mechanisms, which may decrease fallibility to cognitive illusions. We propose that this connection offers a promising field for finding robust effects of bilingualism to cognition. The present experiment puts this idea to test, aiming to address the question of whether bilingualism can decrease susceptibility to manipulative discourse.

The developed task taps into the acceptability of grammatical illusions (2). The task involves both an inhibitory control component and a grammatical processing component in a two-step process. The first step is the suppression of the automatic answer. If the automatic answer is suppressed, the second step involves computing different grammar-based interpretations; a process that has been evidenced in the reaction times of previous experiments [27, 29]. The peculiarity of the tested stimuli is that it appears well-formed, through blending two locally coherent templates, comparison of events and comparison of individuals, in a way that does not involve any *marked* deviation from what the parser expects to encounter in a well-formed comparative structure [29, 39]. In this sense, the task approaches bilingual linguistic processing in a novel way, because it differs from previous research that found an enhanced bilingual performance in acceptability tasks that tested anomalies such as 'If I am sick again tomorrow, I will have to see my fireman' [40, 41]. What differentiates the two types of stimuli is that in the latter example, the stimulus itself provides a salient cue that calls for inhibition (i.e., 'fireman'). In the grammatical illusions, however, the anomaly is hidden in grammar and the stimulus looks meaningful. In fact, even after completing the task, it is hard to convince some participants that the illusions are ill-formed and lack meaning. This happens because there is no salient cue in the stimulus that facilitates the inhibition of the automatic assumption that the sentence *has* meaning.

## Methods

### Procedure

A timed task was run in Ibex Farm [42], collecting two measures: (i) acceptability judgments on a 3-point Likert scale that featured the options 'correct', 'neither correct nor wrong', and

'wrong' and (ii) reaction times. Participants (N = 276) received the test stimuli one by one, in the form of written sentences, and were asked to press a key, judging their well-formedness on the 3-point scale. They were not given the option to skip a sentence or go back and change a previously given answer.

Before the actual experiment started, a warm-up session familiarized the participants with placing three fingers on the keys '1' (wrong), '2' (neither correct nor wrong), and '3' (correct) in order to select an answer without using a mouse, thus minimizing noise in the online measure. Two keys were assigned to the dominant hand. The order of presentation of the sentences was pseudo-randomized across participants.

## Participants

All participants were neurotypical adults capable of providing informed consent. They all provided written consent prior to participation in the study in accordance with the Declaration of Helsinki. The Norwegian Centre for Research Data screened and approved the experiment (number: 55775/3/LH).

The participants fell into two groups: 138 monolingual speakers of Greek (73 female, average age: 37.2, SD: 11.6) and 138 bilingual speakers of Greek and a Germanic language, mainly English, German, Norwegian, Swedish, or Danish (73 female, average age: 38.5, SD: 9.4). A design with a sample size of 138 in each group can detect effect sizes of $\delta \geq 0.5$ with a probability of at least 0.985, assuming a two-sided criterion for detection that allows for a maximum Type I error rate of a = 0.05.

The two groups were matched across indices (i.e., age, gender, education). The language of testing was Greek. The bilingual participants were raised monolingually in Greece and only relocated to another country as adults (mainly in Scandinavia, the UK or Germany). All bilingual participants reported having Greek as their L1 and some knowledge of at least one Germanic language (mainly English, German and/or Norwegian) to a degree that ranged from very good to near-native. Moreover, all bilingual participants were living outside of Greece at the time of testing. The lower limit of the period of residence outside Greece was set to 4 years (average: 11.9, SD: 9.9), in order to establish prolonged periods of L1 inhibition and exposure to an L2/3. There was no upper limit, but the obtained maximum was 47 years. All monolingual participants reported that they had not lived outside Greece for more than one year, and the vast majority of them had never lived abroad, thus ensuring the absence of a period of L1 inhibition and/or dense language-switching experience in the monolingual group. Most of the monolingual participants reported some basic or intermediate knowledge of English, but no actual use on a frequent basis. The demographic profile for each participant is available in the following open access repository: http://repositori.urv.cat/fourrepopublic/search/item/PC%3A3577. All participants were recruited online, mainly through responding to invitations posted on social media platforms, and completed the research online, in Ibex Farm. They were recruited through advertisements posted in the social media of Greek communities in Scandinavia, the UK, and Germany. Participant exclusion criteria included non-native knowledge of Greek, reception of speech-pathology treatment, presence of neurological disorders (all three assessed on the basis of self-report), and inappropriate behavior with the fillers in the task (i.e., a series of non-target answers). On the basis of these criteria, 18 participants (not counted in the two groups of 276 people mentioned above) were not included in the study.

## Task design

The task involves items that fall into two conditions, illusions vs. fillers, with the latter acting as a control for obtaining baseline comparisons. Each sentence (i.e., both illusions and fillers)

is 21 syllables long after phonetic transcription. Both grammatical and ungrammatical fillers were used. All 276 participants encountered all test items. This means that, excluding fillers, 5520 data points were collected, 2760 for each measure (i.e., acceptability judgments and reaction times). The complete task together with the obtained dataset is available in the following open access repository: http://repositori.urv.cat/fourrepopublic/search/item/PC%3A3577.

The test items in the illusion condition feature comparative illusions that fall into two categories: category 1 involves a repeatable predicate (3), while category 2 features a non-repeatable predicate (4).

3. Perisoteri    anθropi exun  pai  sto   Lonðino ap'  o,ti   eγo.
   More          people  have  been to the London from PRON I.NOM
   'More people have been to London than I have.'

4. Perisoteri    adres exun  teʎosi to    sxolio ap'  o,ti   aftos.
   More          men   have  finished the school from PRON he.NOM
   'More men have finished school than he has.'

The use of two categories is motivated by previous reports of repeatability effects for English, and more specifically, by the claim that repeatable predicates induce higher ratings of the illusions [26]. However, preliminary research on the semantics of the stimuli we used in the present experiment show that this effect of repeatability does not exist in Greek [29], hence we do not elaborate on the notion of repeatability in the next sections. Following recommendations for experimental design in previous literature [43], the task was designed to feature a 2:1 ratio of fillers to test items and a 1:1 ratio of acceptable to unacceptable items, taking fillers and test items together. This translates into 10 test items in the illusion condition and 20 items in the filler condition. The fillers consist of 15 ungrammatical and 5 grammatical sentences. The reason for this is the following: Ungrammatical test items would typically be classified as unacceptable, but the tested stimuli are a good example of the dissociation between grammaticality and acceptability [44]: Even though the illusions are ungrammatical, previous research has shown them to be highly acceptable [26, 27], hence in terms of the experimental design, we calculated them as such. This means that, in order to achieve a 1:1 ratio of acceptable and unacceptable items in total, the ratio of ungrammatical fillers to grammatical fillers had to be 3:1.

The sensitivity of the task in terms of measuring fallibility to illusions in the target language was piloted and successfully demonstrated in previous research with 30 monolingual speakers of Greek (included in the monolingual control group of the present study) and 30 bidialectal speakers of Greek and Cypriot Greek [29]. At present, this is the only study that has addressed grammatical illusions from a comparative perspective, by testing and comparing two populations, one monolingual and one bidialectal (i.e., speaking two varieties of the same language). The results showed a better performance of the bidialectal group in terms of rejecting the illusions as ill-formed, but this finding could not be unambiguously attributed to an effect of linguistic background, as several other factors might have played a role; for instance, particularities of the sociolinguistic profile of the tested population, which may affect judgments in the standard variety [29].

## Results

Decisions with reaction times slower than 2.5 SD of the mean of each experimental condition were trimmed as outliers. This resulted in the exclusion of approximately 0,4% of the data (including the corresponding judgements) in the illusion trials and 1% of the data in the filler trials. Fig 2 presents the distribution of raw acceptability judgments for the two groups of participants in illusions.

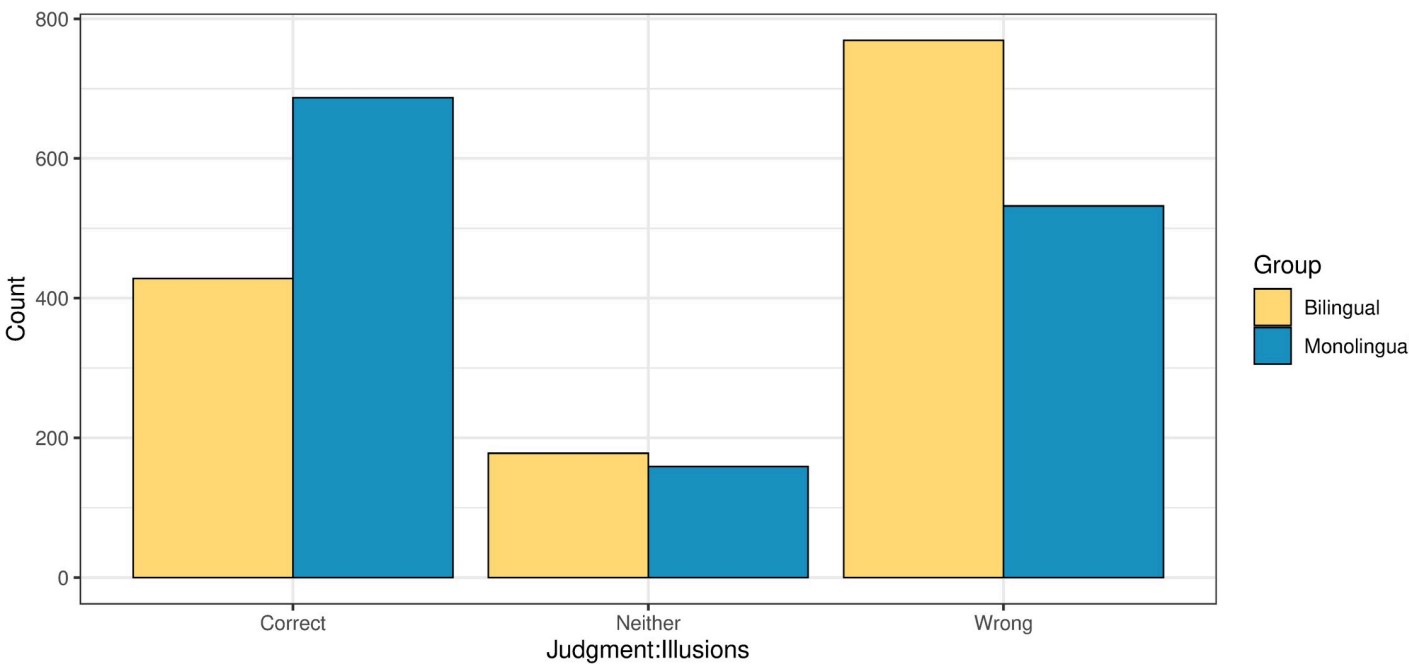

**Fig 2. Acceptability judgments across the two groups in illusions.**

The number of times when the participants responded 'neither correct nor wrong' was comparable for the two groups (327 vs. 275 responses for the bilinguals and monolinguals respectively, taking into account both illusions and fillers). A generalized linear mixed effects model analysis revealed no significant differences in the probability of 'neither' responses between the two groups of participants overall, and across individual conditions (p = 0.27 in the filler trials, p = 0.54 in the illusion trials).

All generalized linear mixed effects models were fit using the lme4 package of the software R version 4.0.2 [45]. Post-hoc pairwise comparisons were run using the R package emmeans [46], with adjusted alpha levels (Tukey's method to control for multiple comparisons). For the subsequent analyses of accuracy, we grouped the 'neither' responses together with the non-target responses (i.e., 'correct' in illusions, 'wrong' in grammatical fillers, 'correct' in ungrammatical fillers), because these responses entail a failure to either unambiguously reject the manipulative stimuli as ill-formed in the illusion condition or provide the target correct/wrong judgment in grammatical and ungrammatical fillers respectively.

Fig 3 illustrates the participants' accuracy in the illusions and the fillers.

As evident from Fig 3, monolingual and bilingual participants are equally accurate in the filler trials, scoring at 83% and 84% respectively. At the same time, bilinguals outperform monolinguals in the illusions, effectively detecting the ill-formedness of the illusions and rejecting them as 'wrong' (56% vs. 39% accuracy). To test this statistically, we ran a binomial generalized linear mixed effects model analysis where accuracy was predicted based on an interaction of two factors: Group (bilingual or monolingual) and condition (filler or illusion). Age was added as a main effect in this analysis of acceptability judgments, and the age factor is also relevant for the analysis of the reaction times, given that older participants may reasonably take longer to respond. Participants and items were included as random effects. The dependent variable in the binomial generalized linear mixed effects model was accuracy, coded as 1 (accurate) or 0 (inaccurate). The independent variables were dummy-coded: the variable *group* had two levels, Bilingual (0) and Monolingual (1); and the variable *condition* had two

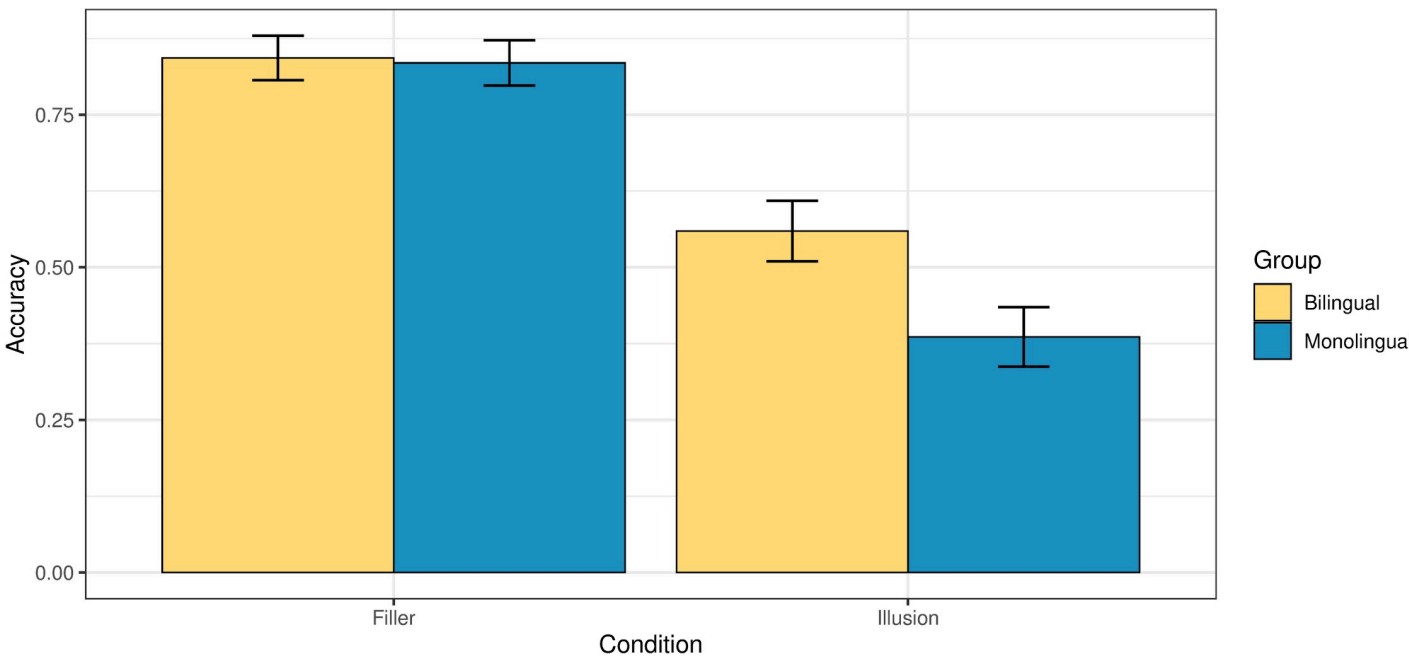

**Fig 3. Accuracy in providing the target judgment by condition and group.**

levels, Filler (0) and Test (1). The model showed a significant effect of condition (i.e., lower accuracy in the illusions, but not in fillers), and a significant interaction of group and condition (p<0.001). The output is given in Table 2a in the S1 File. Post-hoc pairwise comparisons (with Tukey correction for multiple comparisons) revealed a highly significant difference in accuracy between monolinguals and bilinguals on the illusion condition (p<0.001), and no significant difference between the groups on the filler condition (p = 0.96; Table 2b in the S1 File). Table 1 summarizes the accuracy of the two groups across the two conditions.

Turning to the online measure, Fig 4 presents the global reaction times associated with the three judgments in whole dataset. Figs 5 and 6 present the reaction times behind the different judgments, separately for illusions and fillers. In all cases, the 'neither' response is associated with the longest reaction times, followed by 'wrong' and then 'correct'. We conducted a linear model analysis where common log reaction time was predicted based on an interaction of judgement type (correct, neither or wrong) and group (bilingual or monolingual). Age was added as a main effect. Taking illusions and fillers together, the effect of judgement type was significant, indicating that participants took significantly more time to think before they responded 'wrong' or 'neither' than 'correct' (p<0.001 for both 'wrong' and 'neither'; Table 3a, 3b in the S1 File).

A separate analysis of the illusion condition also revealed a significant effect of judgement type. Participants were significantly slower when giving to illusions the judgment 'wrong' or

**Table 1. Accuracy across the two groups.**

| Group | Condition | Accuracy | SE |
|---|---|---|---|
| Bilingual | Filler | 0.84 | 0.04 |
| Bilingual | Illusion | 0.56 | 0.05 |
| Monolingual | Filler | 0.83 | 0.04 |
| Monolingual | Illusion | 0.39 | 0.05 |

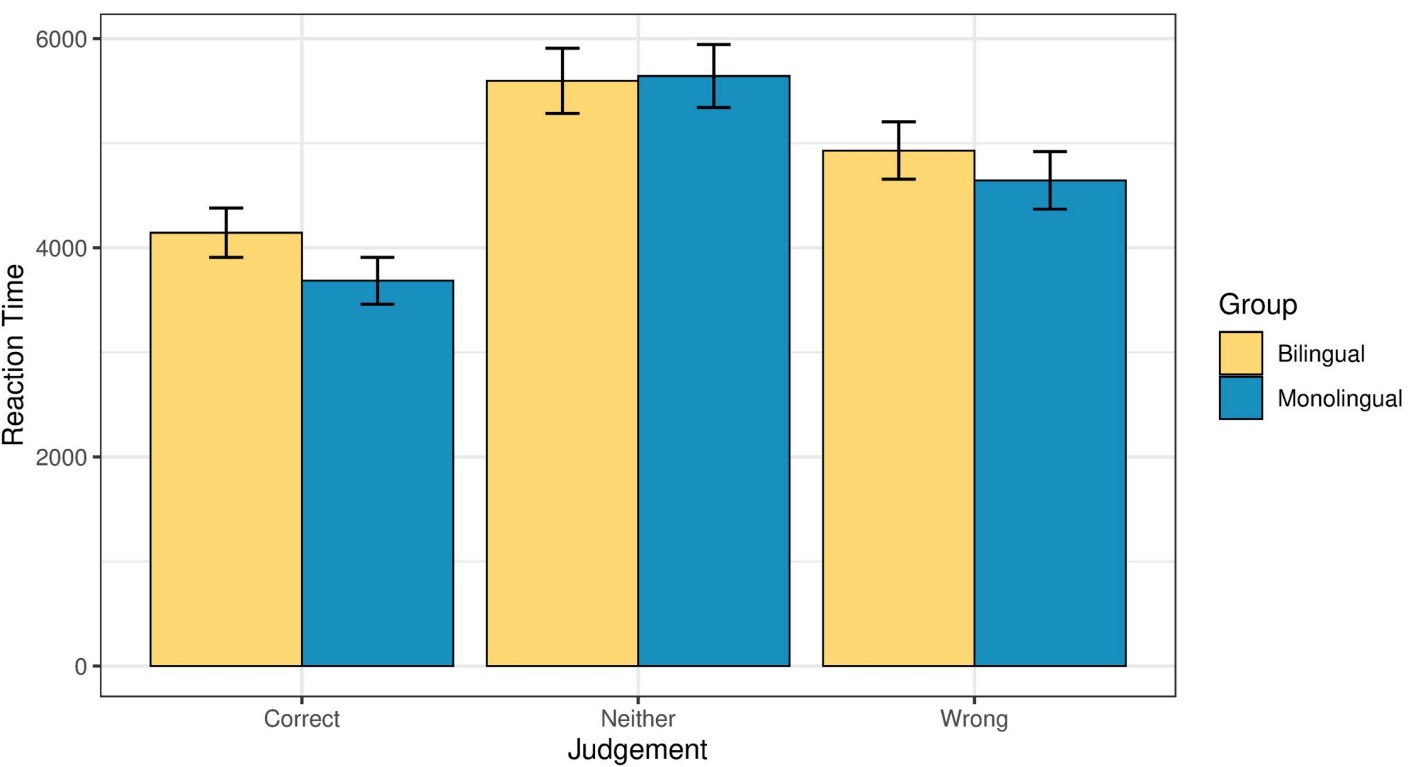

**Fig 4. Global reaction times (illusions and fillers together) across the two groups.** The y axis shows milliseconds (ms).

'neither' than when they accepted them as 'correct' (p<0.001 in both comparisons; Table 4a in the S1 File). This finding is in line with the reaction times reported for illusions in previous literature, where it was documented that people are fast at getting it wrong [27].

The main effect of age was not statistically significant (p = 0.25). The effect of group was significant, when taking illusions and fillers together, with the bilingual participants being slower than the monolinguals (p<0.001, Table 3a in the S1 File). Post-hoc pairwise comparisons (given in Table 3b in the S1 File) revealed that the bilinguals were significantly slower than the monolinguals when they gave the judgment 'correct' or 'wrong' (p<0.001 in both cases), with no difference in the reaction times between the groups when they responded 'neither' (p = 0.53).

The separate analysis of the illusion condition also revealed a significant effect of group: the monolinguals responded significantly faster than the bilinguals (p<0.001; Table 4a in the S1 File). Post-hoc pairwise comparisons (Table 4b in the S1 File) showed that the bilinguals were significantly slower than the monolinguals when they responded 'correct' (p = 0.002), marginally but not significantly slower when they responded 'wrong' (p = 0.07), and with a numerical trend in the predicted direction, but no significant difference when they responded 'neither' (p = 0.68).

In relation to the reaction times, we ran an additional model, where the 'neither' responses were excluded. The model included a three-way interaction of group, condition, and judgement type, with age added as a main effect. The model revealed three significant main effects: Group (bilinguals were significantly slower than the monolinguals, p<0.001), condition (both groups were significantly slower on the filler condition than on illusions, p<0.001), and judgement type (participants from both groups were significantly slower when they judged the sentence as 'wrong' than when they judged it as 'correct', p<0.001). The effect of age was not

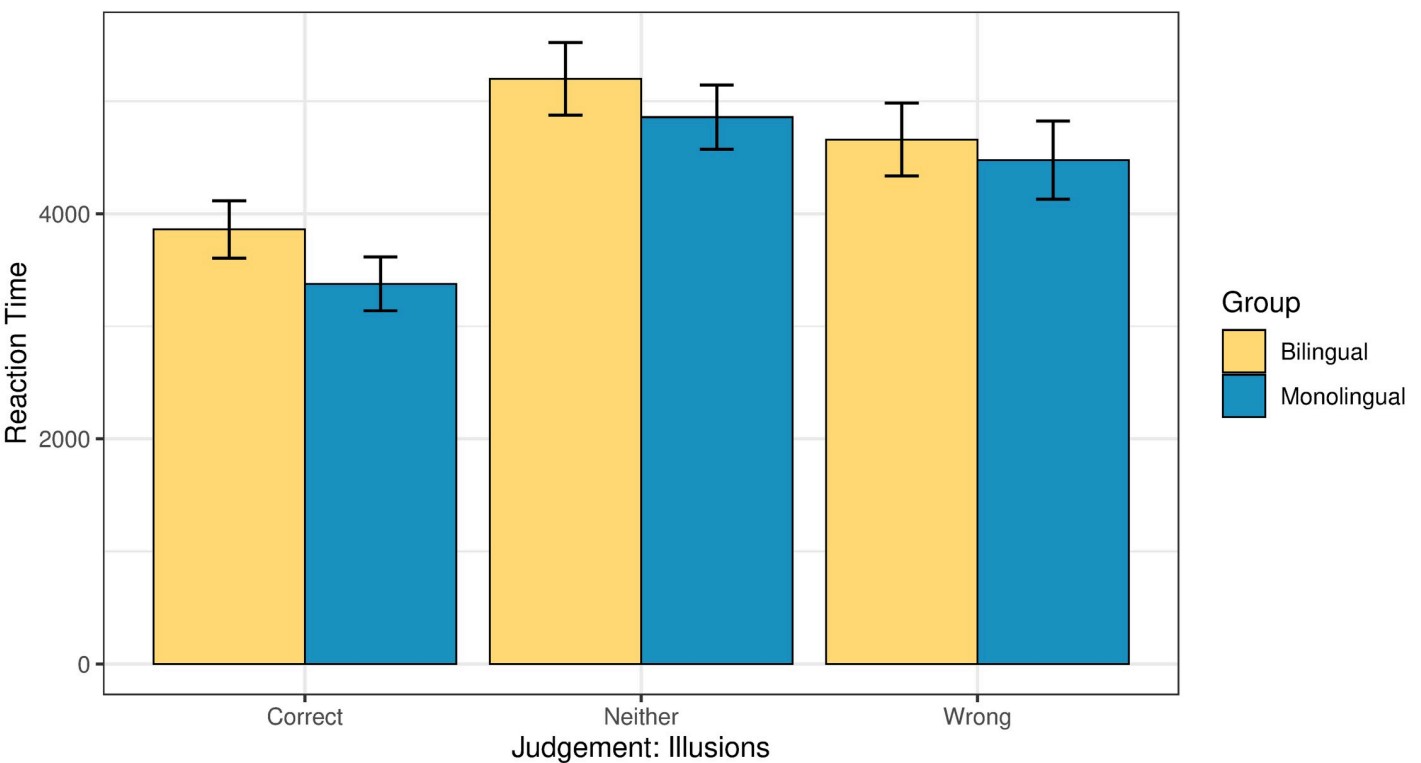

**Fig 5. Reaction times (ms) split for judgment type in illusions across the two groups.**

significant, and no significant interactions were observed. The output of the model is given in Tables 5a, 5b in the S1 File. Post-hoc pairwise comparisons revealed that bilinguals were numerically slower than the monolinguals in the illusion condition when they judged the sentences as 'wrong' (i.e., the target answer), but the difference did not reach statistical significance (p = 0.17).

Zooming in on the performance of the bilingual group, we observed that the length of bilingual experience was a significant predictor of the ability to detect the illusions. We split the bilinguals into two groups by the median based on the number of years they spent in a country where their L1 is not spoken. Participants who spent more than 7 years in a context that required L1 suppression formed the "long bilingual experience" group (68 participants, average age: 44.2). Participants with 7 or less years of bilingual experience were grouped into the "short bilingual experience" group (70 participants, average age: 33.1). Monolinguals with no or scarce experience of L1 suppression counted as a separate group (138 participants). Fig 7 illustrates the accuracy in illusions and fillers for the three groups of participants. All three perform remarkably similar on providing the target response in the filler trials, however, there is substantial diversity in their accuracy in detecting the illusions, going from 39% in the monolingual group to 62% in the "short bilingual experience" group and then down to 49% in the "long bilingual experience" group. Table 2 summarizes the accuracy of the three groups across the two conditions.

We fit a generalized linear mixed model where accuracy was predicted based on an interaction of group (short bilingual experience, long bilingual experience, or monolingual) and condition (illusion or filler). Participants and items were included as random effects. The model

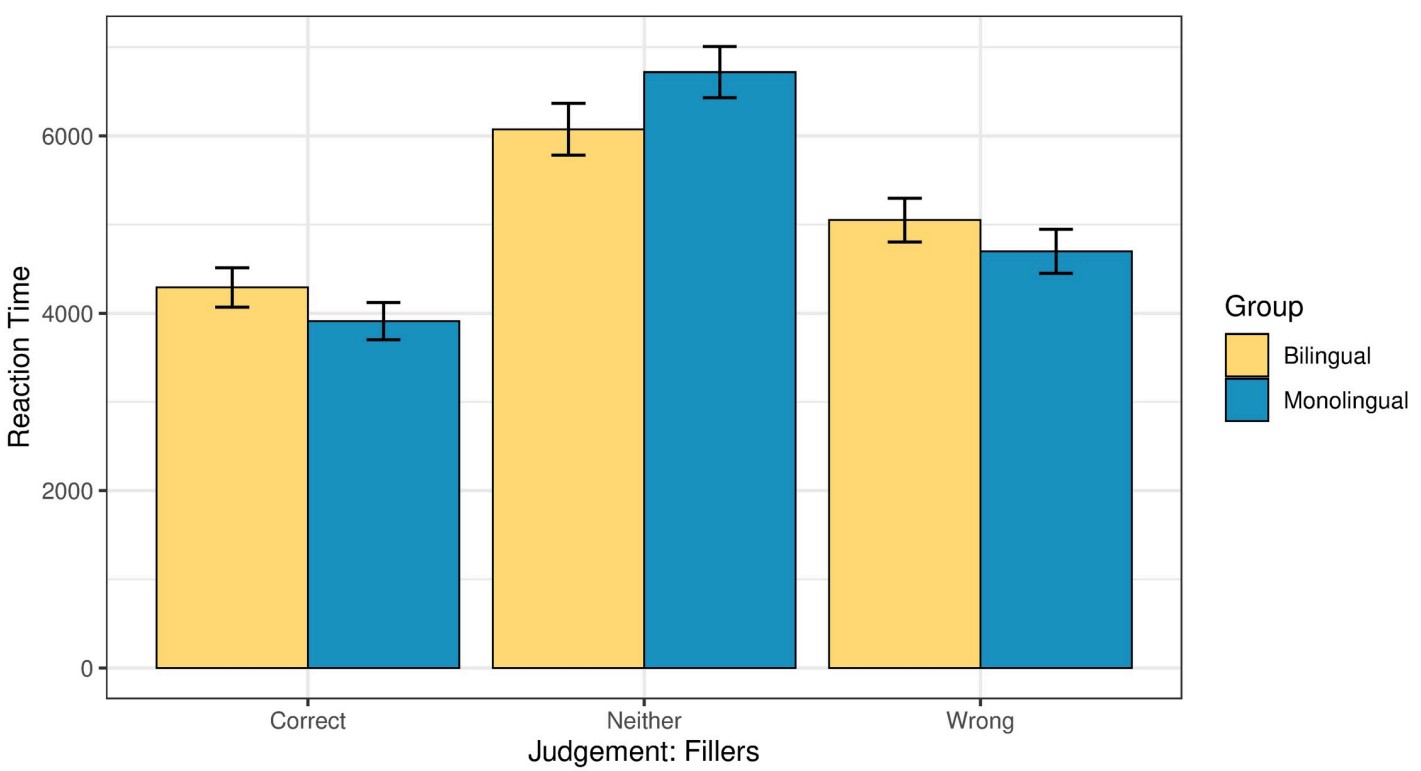

**Fig 6. Reaction times (ms) split for judgment type in fillers across the two groups.**

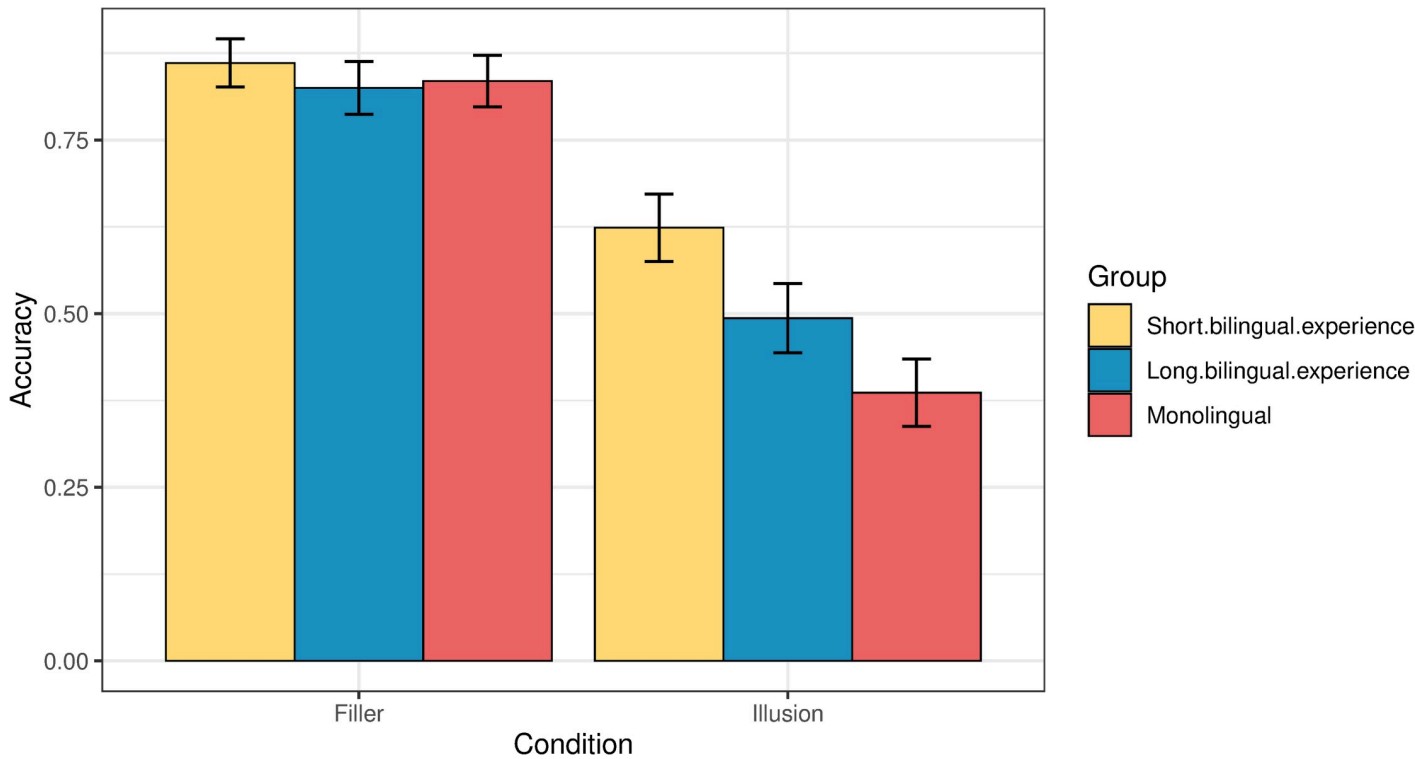

**Fig 7. Accuracy in giving the target judgment across the three groups.**

**Table 2. Accuracy across the three groups.**

| Group | Condition | Accuracy | SE |
|---|---|---|---|
| Short bilingual experience | Filler | 0.86 | 0.03 |
| Short bilingual experience | Illusion | 0.62 | 0.05 |
| Long bilingual experience | Filler | 0.82 | 0.04 |
| Long bilingual experience | Illusion | 0.49 | 0.05 |
| Monolingual | Filler | 0.83 | 0.04 |
| Monolingual | Illusion | 0.39 | 0.05 |

revealed a significant effect of condition: all three groups were significantly less accurate on the illusions than on fillers (p<0.001). The output of the model is given in Table 6a in the S1 File. Post-hoc pairwise comparisons revealed no significant differences between the three groups on the filler trials. At the same time, we found that all groups were significantly different from each other in detecting the illusions. Participants with short bilingual experience ($\leq$ 7 years) performed significantly more accurately than the monolinguals (p<0.001). Participants with long bilingual exposure (> 7 years) also performed significantly better than the monolinguals (p = 0.03), however, they were also significantly less accurate than the bilinguals with a shorter L2 experience (p = 0.04).

With respect to the role of the length of bilingual experience in relation to the reaction times, Fig 8 shows the performance of the three groups. The monolinguals are the fastest ones to judge the stimuli across the board, both in illusion and filler trials, followed by bilinguals with long bilingual experience, and then by bilinguals with short bilingual experience. To

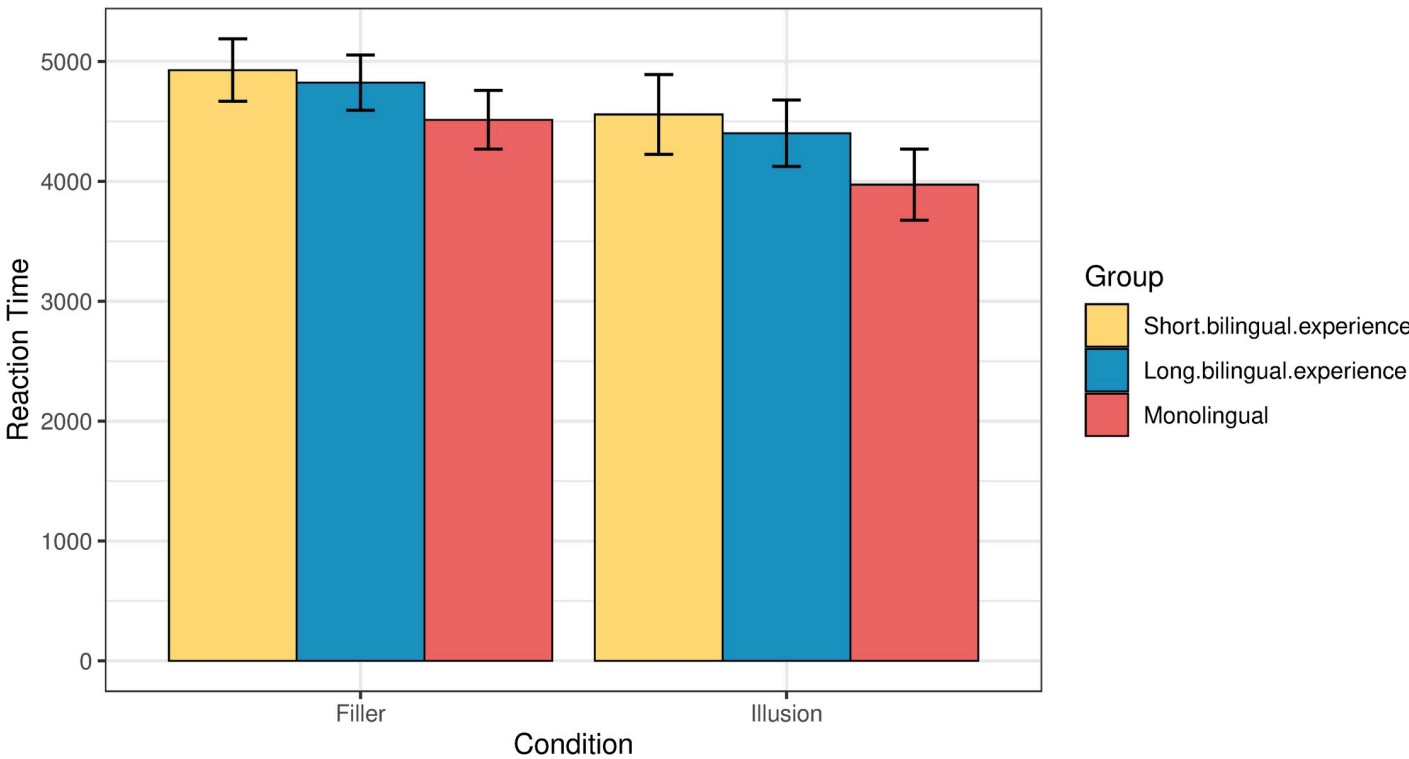

**Fig 8. Reaction times (ms) by condition across the three groups.**

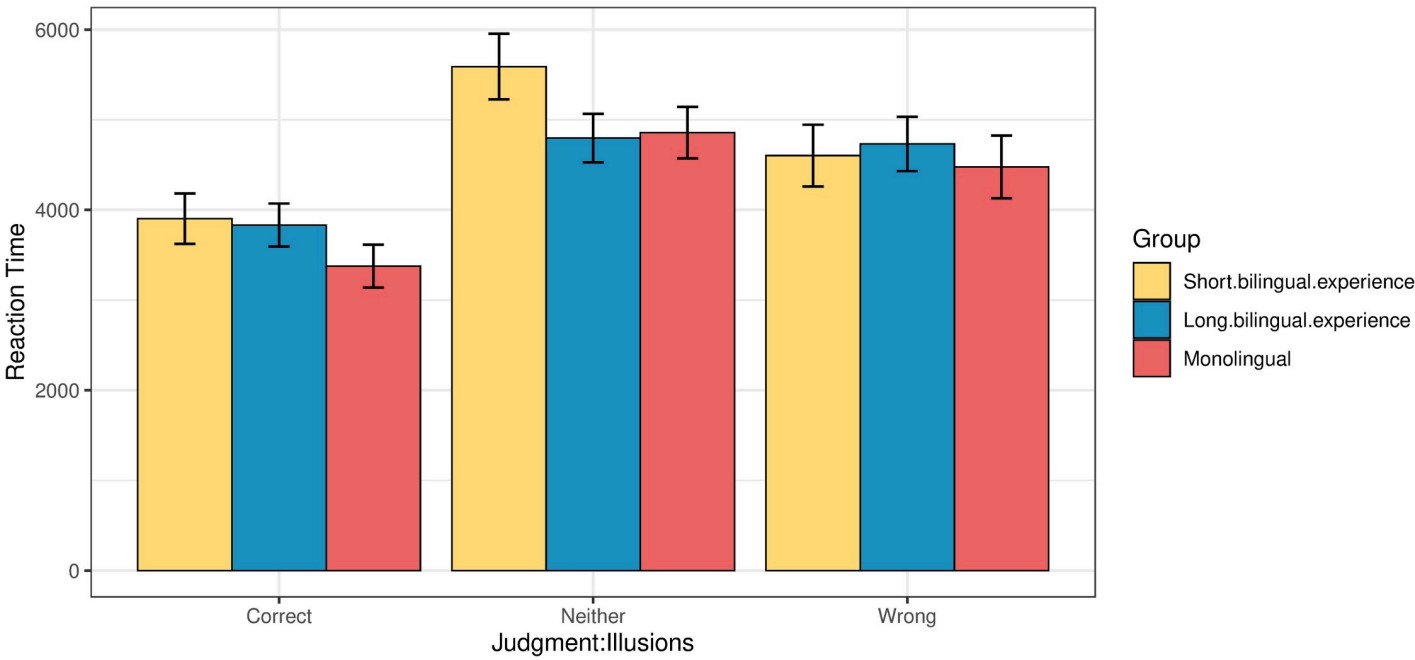

**Fig 9. Reaction times (ms) split for judgment in illusions across the three groups.**

analyze the results statistically, we fit a linear model predicting the common log reaction time as an interaction of condition and group, with age added as a separate main effect. The contrasts between the reaction times of the two bilingual groups and the monolinguals were significant in both conditions, illusions and fillers (p<0.001 for all contrasts). At the same time, the differences between the two bilingual groups did not reach significance (p = 0.99 for both contrasts). The output of the model is given in Tables 7a, 7b in the S1 File.

As mentioned already, age is an expected covariate with length of language use. However, the results did not associate the oldest subgroup with longer reaction times. The group of bilinguals with long bilingual experience is significantly older than the group with short bilingual experience (Levene's Test for equality of variance F = 10.997, p = 0.001), but, as Fig 8 shows, they responded faster than the younger bilingual group, in both illusions and fillers.

Fig 9 shows the association between reaction times and judgments in illusions across the three response types. A logistic regression model analysis was run, where the log reaction time was predicted as an interaction of group and judgement type, with age as additional fixed effect. Post-hoc pairwise comparisons indicated that both bilingual groups were significantly slower than the monolinguals when they responded 'correct' (monolinguals vs. long bilingual experience: p = 0.007; monolinguals vs. short bilingual experience: p = 0.01), while there was no statistically significant difference between the two bilingual groups for the responses of this type (p = 0.99). No statistically significant differences were found between the groups when they responded 'neither' (monolinguals vs. long bilingual experience: p = 0.75; monolinguals vs. short bilingual experience: p = 0.35; short vs. long bilingual experience: p = 0.15). Bilinguals with long bilingual experience were marginally slower than the monolinguals when they responded 'wrong' (p = 0.08), while bilinguals with short bilingual experience were only numerically, but not statistically, slower than the monolinguals (p = 0.57), and there was no statistically significant difference between the two bilingual groups for the responses of this type (p = 0.5; see Tables 8a, 8b in the S1 File).

## Discussion

This experiment taps into the potentially differential susceptibility to manipulative discourse in a monolingual and a bilingual population. The results reveal a novel trade-off in the domain of language processing: (i) bilinguals are more accurate than monolinguals in spotting and rejecting tested comparative illusions, but (ii) they are also slower across the board in judging the linguistic stimuli they are presented with.

Comparative illusions are an excellent vehicle to test whether bilinguals/plurilinguals rely less on automatic answers when processing linguistic stimuli. Previous research has shown that having two languages may translate into an enhanced ability to attend selectively to various aspects of the linguistic stimuli, (dis)engaging attentional resources accordingly such that the illusory effect is detected and the illusion is rejected as an ill-formed utterance [29]. Although this idea seems to invoke the notion of a bilingual advantage, which is compatible with our results too, we suggest that it provides only half of the picture. If a robust (dis)advantageous effect in any domain exists, it necessarily forms part of a *trade-off*: enhanced performance in one cognitive measure/domain must be compensated by decreased accuracy and/or processing speed in another measure/domain. In biology, such trade-offs are frequent and emerge because one trait cannot be optimized without conferring an expense on other traits, given that organisms function together as integrated wholes in the Darwinian sense [47]. In this context, we frame our results through what we refer to as the *Plurilingual Adaptive Trade-off Hypothesis* (PATH).

Building on Green & Abutalebi's *Adaptive Control Hypothesis*, which suggests that bilinguals' interactional contexts entail an adaptive alteration of their language control abilities [48], PATH proposes that (i) this adaptive alteration may result in enhanced pragmatic monitoring that decreases fallibility to manipulative discourse by means of recruiting sharpened top-down control processes, while (ii) adding the crucial notion of a trade-off. The latter should be understood as a fitness trade-off, referring to a negative correlation between processes and traits that make use the same finite resources, such that they cannot all be optimized at the same time [49]. Although the results obtained from the present experiment anchor PATH specifically to the ability to detect manipulative discourse, a larger bundle of effects can be subsumed under PATH. For example, the enhanced pragmatic monitoring may be deriving from a more general ability for sharpened error-monitoring, which has been linked also to metacognitive processing [50]. According to PATH, a cognitive alteration that derives from bilinguals' conversational experiences confers a bundle of effects in the domain of language processing, but not a bilingual advantage or a bilingual disadvantage per se. The shift is from unitary concepts that imply a positive or negative value to mosaic trade-offs that throughout the lifespan remain plastic, adaptable, and sensitive to the continuously changing individual experiences that are associated with the linguistic developmental trajectory. Previous studies that have found an enhanced error-monitoring ability in the linguistic domain for bilinguals also recognized the effect of individual experience [50]. This effect explains the role of length of bilingual experience in our findings. Upon splitting the bilingual group into two groups by the median based on the length of bilingual experience (>7 years vs. ≤7 years), we found that bilinguals with a long bilingual experience performed significantly worse in detecting the manipulative stimuli than their peers who had a short bilingual experience. Crucially, they were not significantly worse in giving the target judgment for grammatical or ungrammatical fillers, so this finding is specific to illusions. At the same time, both groups of bilinguals were still significantly better than their monolingual peers in spotting and rejecting the illusions. Thus, there is a tripartite distinction in the data: participants with a short bilingual experience perform best at spotting the illusions, followed by participants with a long bilingual experience,

and finally participants with no/extremely scarce bilingual experience, who are effectively classified as monolinguals. This picture recognizes subtler patterns of variance within the groups and is in line with recent calls to view bilingualism and plurilingualism as nuanced spectrum experiences rather as dichotomous categories [51, 52].

The reaction times we obtained also reveal an effect of developmental trajectory that shows within-group variance. This provides support to the claim that the observed trade-off is plastic. We found that a longer bilingual experience (>7 years) lead to a decreased ability to spot illusions compared to a shorter bilingual experience, but the overall reaction times of the former group are shorter compared to those of the latter group (see Figs 7 and 9). This regression to the mean in both measures maintains the benefit-cost equilibrium (i.e., worse in the offline measure, but faster in the online measure) and attests to the nature of the trade-off as a dynamic phenomenon that is sensitive to the actual bilingual experience and the way the latter fluctuates throughout the lifespan [9]. This fluctuation explains why the group with a longer bilingual experience is worse at spotting the illusions, but also faster in providing a judgment, compared to the other bilingual subgroup. Their use of two or more languages progressed from being a process that at the onset of bilingualism relied heavily on executive control to a far more automated switching process that is less and less reliant on top-down control mechanisms, hence the regression to the prior state [53].

Focusing on the main finding of this experiment, the differential performance of monolinguals and bilinguals in detecting the illusions is informative about the granularity of the tests that are used to approach the effects of bilingualism on cognition. In relation to our stimuli, an enhanced performance of bilinguals is observed in the tested phenomenon of comparative illusions, but not in the ungrammatical fillers, which are grammatically ill-formed too. More specifically, the ungrammatical fillers involve either number agreement errors (e.g., 'The key to these cabinets are on the marble table') or morphological errors associated with thematic role reversals (e.g., 'The students that the teacher expelled complained to them'—instead of 'complained to him/her'). Although one could have hypothesized that the bilinguals' enhanced ability for linguistic monitoring could translate into better performance in these sentences too, our results show that this is not the case. The reason for this may have to do with issues of task sensitivity and granularity. Both monolinguals and bilinguals performed very well in the task of rejecting the ungrammatical fillers as ill-formed, hence the slightly better performance of the bilinguals does not reach statistical significance. As mentioned in the Introduction, one of the factors that may explain the contradictory evidence for and against bilingual effects in EFs has to do precisely with issues of task-granularity in this domain [9]: Put simply, not all tasks permit capturing the trade-off effects that bilingualism confers to cognition because (i) the performance of the control group may be sufficiently good, such that any difference is marginal or insignificant from a statistical point of view, and (ii) different measures must be collected in order to form observations about gain-compensation balances. This claim is in line with the studies that find speed-accuracy trade-offs in the EF domain for some tasks (e.g., the Simon task but not the flanker task) and in some populations (e.g., younger bilingual children but not older bilingual children) [54]. An efficiency-oriented trade-off can also explain the better bilingual performance in tasks that test semantic anomalies [40, 41], alongside the lower accuracy of bilinguals in tasks that measure lexical fluency [14]. These so-called 'advantages' or 'disadvantages' are interrelated effects that stem from overall differences in development and processing. These effects may vary across populations, such that it is possible that not all types of bilinguals perform better than monolinguals in all tasks that tap into semantic anomalies or grammatical illusions, but in terms of their general occurrence, these trade-offs necessarily form part of cognition across populations in some form or other (e.g., speed vs. accuracy, efficiency vs. flexibility etc.). In sum, trade-offs are ubiquitous in cognition and in all goal-directed

systems, because enhanced computational performance in any measure/domain does not come for free [49, 55]. Differences in monolingual and bilingual language processing are another field of research that offers evidence for such compensatory ramifications.

Our proposal that the concept of the trade-off is inherent to any bundle of effects that bilingualism confers on cognition has important implications also for recent meta-analyses that claim absence of evidence for a bilingual advantage, but presence of evidence for a bilingual disadvantage [56]. According to PATH, if a disadvantage exists, an advantage is bound to exist in some other measure(s). Finding it is of course a complex matter that requires not only the right testing domain or domain-selection criteria in the case of meta-analyses (e.g., EFs, cognitive illusions, lexical fluency etc.), but also the right stimuli, as evidenced by the similar performance of bilinguals and monolinguals in our ungrammatical fillers. Crucially, the connection we put forth in the present work between sharpened top-down control processes and decreased fallibility to linguistic illusions has the benefit of searching for the bilingual advantage (as part of a trade-off) outside the field of EFs, where research has yielded conflicting results. It is likely that further research on linguistic illusions will determine more nuanced aspects of the conferred effects of bilingualism to neurocognition. For example, one question that the present experiment did not address concerns whether the results of the claimed effects extend to manipulative discourse more broadly, outside the context of linguistic illusions. Another question has to do with the exact background/demographic variables that lead to robust bilingual effects in language processing. This experiment did not measure variables such as degree of proficiency, language proximity, or density and cause of language-switching, but future studies on grammatical processing in bilinguals will likely tap into them and provide insights as to what clusters of factors confer the observed effects.

## Supporting information

**S1 File.** 1a. Probability of giving a 'neither correct nor wrong' response by condition and group. 1b. 'Neither' responses: post-hoc pairwise comparisons. 2a. Accuracy as predicted by an interaction of condition and group. 2b. Accuracy: post-hoc pairwise comparisons. 3a. Reaction times (fillers and illusions together) as predicted by an interaction of judgement type and group, with age as a main effect. 3b. Reaction times (fillers and illusions together): post-hoc pairwise comparisons. 4a. Reaction times (illusions) as predicted by an interaction of judgement type and group, age as a main effect. 4b. Reaction times (illusions): post-hoc pairwise comparisons. 5a. Reaction times (fillers and illusions together, 'neither' responses excluded) as predicted by an interaction of judgement type, condition and group, with age as a main effect. 5b. Reaction times (fillers and illusions together, 'neither' responses excluded): post-hoc pairwise comparisons. 6a. Accuracy as predicted by an interaction of condition and length of bilingual experience group. 6b. Accuracy: post-hoc pairwise comparisons. 7a. Reaction times (fillers and illusions together) as predicted by as an interaction of condition and length of bilingual experience group, with age as a main effect. 7b. Reaction times: post-hoc pairwise comparisons. 8a. Reaction times (illusions) as predicted by an interaction of condition and length of bilingual experience group, with age as a main effect. 8b. Reaction times (illusions): post-hoc pairwise comparisons.
(DOCX)

## Author Contributions

**Conceptualization:** Evelina Leivada.

**Data curation:** Evelina Leivada.

**Formal analysis:** Natalia Mitrofanova.

**Funding acquisition:** Evelina Leivada.

**Investigation:** Evelina Leivada.

**Methodology:** Evelina Leivada.

**Project administration:** Evelina Leivada.

**Supervision:** Marit Westergaard.

**Writing – original draft:** Evelina Leivada, Natalia Mitrofanova.

**Writing – review & editing:** Marit Westergaard.

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
