## [Decision Letter · Decision Letter 0]

19 May 2021

PONE-D-21-10732

Bilinguals are better than monolinguals in detecting manipulative discourse

PLOS ONE

Dear Dr. Leivada,

Thank you for submitting your manuscript to PLOS ONE. After careful consideration, we feel that it has merit but it reauires minor revisions for full aceptance. Therefore, we invite you to submit a revised version of the manuscript that addresses the points raised during the review process.

We look forward to receiving your revised manuscript.

Kind regards,

Itziar Laka, PhD

Academic Editor

PLOS ONE

Journal Requirements:

3. In order to improve reporting, in your methods section, please provide additional information about the participant recruitment method and the demographic details of your participants, such as a table of relevant demographic details, b) a description of how participants were recruited, and c) descriptions of where participants were recruited and where the research took place.

Reviewers' comments:

Reviewer's Responses to Questions

**Comments to the Author**

1. Is the manuscript technically sound, and do the data support the conclusions?

Reviewer #1: Yes

Reviewer #2: Yes

2. Has the statistical analysis been performed appropriately and rigorously? 

Reviewer #1: Yes

Reviewer #2: Yes

3. Have the authors made all data underlying the findings in their manuscript fully available?

Reviewer #1: Yes

Reviewer #2: Yes

4. Is the manuscript presented in an intelligible fashion and written in standard English?

Reviewer #1: Yes

Reviewer #2: Yes

5. Review Comments to the Author

Reviewer #1: The paper investigates differences between bilinguals and monolinguals in the processing of grammatical illusions. Bilinguals were more accurate but slower than monolinguals detecting grammatical illusions. In addition, there was a relationship between years of bilingual experience and the detection of illusions. Bilinguals with lower experience (<7 years) were the more accurate group, followed by those bilinguals with much experience (> 7 years), which in turn were more accurate than monolinguals.

The paper reports a novel effect of bilingualism in detecting manipulative discourse and provides an interesting account for the results obtained. I have some comments.

The introduction is completely framed within the bilingual advantage in EF. The authors justify this choice based on the observation that bilinguals have a better language inhibitory abilities than monolinguals. However, the task employed is within the linguistic domain, where it has been repeatedly reported a bilingual disadvantage, mainly in the speed of processing. It should be better described how the task employed measures the language control and processing at the same time. In the experiment, participants are asked to indicate if a given sentence is grammatically correct or not. So, the correct response for the illusions is “wrong”. Thus, accuracy differences between groups then indicate that bilinguals were more accurate indicating that a sentence was ungrammatical despite being acceptable. Are those results indicative that acceptability is a more automatic judgement and it needs to be suppressed in order to answer to the grammatically judgement?

The results are interpreted in the realm of a trade-off accuracy, but applying to the bilingual experience rather than to the task itself. Could not be the case that this task is a two-step processing entailing a suppression of the automatic response followed by a grammaticality judgement? And that only those participants that accurately suppress the acceptability judgement (the sentence is grammatically acceptable), have a second step, slower, where the grammatical judgement has to be made?

Aside note, if as suggested in the discussion, the PATH proposal stems from a negative correlation between accuracy and RTs, why this is not included.

The method section is too convoluted making very difficult to know what the exact design was. Information regarding the participants, materials and design is mixed. Subsections of methods including participants, design, procedure and data analysis would help to have this information organized.

Regarding the design (page 5, end of second paragraph), as it is written now, it is very difficult to follow. There were 10 experimental sentences, all of them ungrammatical. So, the sentence “15 ungrammatical and 5 grammatical” only applies to the fillers. Is this correct? Likewise, it says: ”This means that, excluding fillers, 5520 data points sere collected, 2760 for each measure”. Which measure? If fillers are excluded and there were 10 illusions, this leads to 2760 data points.

My major concern with these data is that conclusions are based on performance for 10 sentences; How stable is the data regarding these sentences? It would be interesting to see the plot including datapoints for participants or items to see behavior at the individual level.

It is unclear why the age factor was included in the analysis. Especially for the comparison of bilinguals and monolinguals, where the two groups did not differ in terms of age. How the model converges without the age factor? Why age and not education? A first analysis should include these factors and test their influence on the accuracy in detecting illusions. Surprisingly, the age factor is not included in the three group comparison, where age differences seem more obvious. A statistical test should indicate if the 3 groups differed in age.

Grammatically judgements are made in Greek, which for many participants living abroad, Greek must become their second language. It should be clarified if differences between bilinguals and monolinguals show the bilingual use of two languages (relative to only one of monolinguals) or processing in L1 in monolinguals vs L2 in bilinguals.

Related to this, page 8 last paragraph starts with “zooming in on the performance of the bilingual group, we observed that the length of bilingual experience correlates with the ability to detect illusions”. No correlation analysis is provided. Is there a reason for splitting bilinguals in two groups instead of considering experience as a continuous variable?

Minor points

A table with means and statistics for the two/three groups should be included.

More information should be added regarding the language experience of the three groups.

It is not mentioned in the text, but I guess reaction times only included correct responses.

There are some typos in the text. For instance, page 4 end second paragraph, to address different monolinguals interlocutors, should be monolingual interlocutors.

Discussion first paragraph. Bilinguals are better than. It would be more appropriate bilinguals are more accurate than.

Reviewer #2: I have provided all the comments in the attached file. I have provided all the comments in the attached file. I have provided all the comments in the attached file. I have provided all the comments in the attached file.

6. PLOS authors have the option to publish the peer review history of their article (what does this mean?). If published, this will include your full peer review and any attached files.

Reviewer #1: No

Reviewer #2: **Yes: **Anna Pineda

---

## [Author Response · Author response to Decision Letter 0]

30 Jun 2021

A separate response letter has been uploaded.

---

## [Decision Letter · Decision Letter 1]

2 Aug 2021

Bilinguals are better than monolinguals in detecting manipulative discourse

PONE-D-21-10732R1

Dear Dr. Leivada,

We’re pleased to inform you that your manuscript has been judged scientifically suitable for publication and will be formally accepted for publication once it meets all outstanding technical requirements.

Kind regards,

Itziar Laka, PhD

Academic Editor

PLOS ONE

Additional Editor Comments (optional):

Reviewers' comments:

Reviewer's Responses to Questions

**Comments to the Author**

1. If the authors have adequately addressed your comments raised in a previous round of review and you feel that this manuscript is now acceptable for publication, you may indicate that here to bypass the “Comments to the Author” section, enter your conflict of interest statement in the “Confidential to Editor” section, and submit your "Accept" recommendation.

Reviewer #1: All comments have been addressed

Reviewer #2: All comments have been addressed

2. Is the manuscript technically sound, and do the data support the conclusions?

Reviewer #1: Yes

Reviewer #2: Yes

3. Has the statistical analysis been performed appropriately and rigorously? 

Reviewer #1: Yes

Reviewer #2: I Don't Know

4. Have the authors made all data underlying the findings in their manuscript fully available?

Reviewer #1: Yes

Reviewer #2: Yes

5. Is the manuscript presented in an intelligible fashion and written in standard English?

Reviewer #1: Yes

Reviewer #2: Yes

6. Review Comments to the Author

Reviewer #1: I thank the authors for their review. All my comments on the previous version of the manuscript have been properly addressed.

Reviewer #2: I am very happy with the revised version of the manuscript and I consider that all my comments were addressed.

7. PLOS authors have the option to publish the peer review history of their article (what does this mean?). If published, this will include your full peer review and any attached files.

Reviewer #1: No

Reviewer #2: **Yes: **Anna Pineda

---

## [Editor Report · Acceptance letter]

16 Aug 2021

PONE-D-21-10732R1 

Bilinguals are better than monolinguals in detecting manipulative discourse 

Dear Dr. Leivada:

I'm pleased to inform you that your manuscript has been deemed suitable for publication in PLOS ONE. Congratulations! Your manuscript is now with our production department. 

Kind regards, 

on behalf of

Dr. Itziar Laka 

Academic Editor

PLOS ONE